# Prognostic roles of diabetes mellitus and hypertension in advanced hepatocellular carcinoma treated with sorafenib

**Ming-Han Hsieh**[1,2], **Tzu-Yu Kao**[2¤a], **Ting-Hui Hsieh**[2¤b], **Chun-Chi Kao**[2], **Cheng-Yuan Peng**[1,2], **Hsueh-Chou Lai**[2], **Po-Heng Chuang**[2], **Jung-Ta Kao**[1,2]\*

**1** Department of Medicine, School of Medicine, China Medical University, Taichung, Taiwan, **2** Division of Gastroenterology and Hepatology, Department of Internal Medicine, China Medical University Hospital, Taichung, Taiwan

¤a Current address: Faculty of Medicine, Wroclaw Medical University, Wroclaw, Poland
¤b Current address: Interdisciplinary Program for Undergraduates, National Yang-Ming University, Taipei, Taiwan
\* garrydarkao@gmail.com

## Abstract

### Background & aims

It remains limited whether diabetes mellitus (DM) and hypertension (HTN) affect the prognosis of advanced hepatocellular carcinoma (HCC) treated with sorafenib. Our study attempted to elucidate the roles of DM/HTN and the effects of diabetes medications among advanced HCC patients receiving sorafenib.

### Methods

From August 2012 to February 2018, 733 advanced HCC patients receiving sorafenib were enrolled at China Medical University, Taichung, Taiwan. According to the presence/absence of DM or HTN, they were divided into four groups: control [DM(-)/HTN(-), n = 353], DM-only [DM(+)/HTN(-), n = 91], HTN-only [DM(-)/HTN(+), n = 184] and DM+HTN groups [DM (+)/HTN(+), n = 105]. Based on the types of diabetes medications, there were three groups among DM patients (the combined cohort of DM-only and DM+HTN groups), including metformin (n = 63), non-metformin oral hypoglycemic agent (OHA) (n = 104) and regular insulin (RI)/neutral protamine hagedorn (NPH) groups (n = 29). We then assessed the survival differences between these groups.

### Results

DM-only and DM+HTN groups significantly presented longer overall survival (OS) than control group (control vs. DM-only, 7.70 vs. 11.83 months, p = 0.003; control vs. DM+HTN, 7.70 vs. 11.43 months, p = 0.008). However, there was no significant OS difference between control and HTN-only group (7.70 vs. 8.80 months, p = 0.111). Besides, all groups of DM patients showed significantly longer OS than control group (control vs. metformin, 7.70 vs. 12.60 months, p = 0.011; control vs. non-metformin OHA, 7.70 vs. 10.80 months, p = 0.016; control vs. RI/NPH, 7.70 vs. 15.20 months, p = 0.026).

**Data Availability Statement:** All relevant data are within the manuscript and its Supporting Information files.

**Funding:** The authors received no specific funding for this work.

**Competing interests:** The authors have declared that no competing interests exist.

## Conclusions

Rather than HTN, DM predicts better prognosis in advanced HCC treated with sorafenib. Besides, metformin, non-metformin OHA and RI/NPH are associated with longer survival among DM-related advanced HCC patients receiving sorafenib.

## Introduction

In 2018, liver cancer is estimated to be the sixth most frequently diagnosed cancer and the fourth prevailing cause of cancer-related deaths globally with approximately 841,000 new cases and 782,000 deaths [1]. Among primary liver cancer, hepatocellular carcinoma (HCC) is the most common type [2]. Although various therapeutic modalities have been applied in HCC management, the mortality of HCC patients is still high due to a large proportion of cases diagnosed with advanced tumors [3]. Patients with advanced stage HCC are defined as those with intra-hepatic venous invasion and/or extra-hepatic metastases but preserved liver function (Child-Pugh class A) [4]. Before the development and approval of targeted therapy, transarterial embolization and conformal radiotherapy were available for advanced HCC and both exerted a survival benefit in comparison with supportive care [5]. At present, systemic therapy is recommended for patients with advanced HCC or well-selected HCC patients with Child-Pugh class B cirrhosis plus intra-hepatic venous invasion and/or metastatic disease [6]. Among systemic therapy for HCC, sorafenib is the standard first-line agent [4]. As a multi-kinase inhibitor, sorafenib targets Raf-1 or B-Raf/MEK/ERK signaling and tyrosine kinases of vascular endothelial growth factor (VEGF) receptor (VEGFR)-1, VEGFR-2, VEGFR-3 and platelet-derived growth factor receptor (PDGFR)-β, thus inhibiting tumor survival, proliferation and angiogenesis [7, 8]. Sorafenib has been validated to improve the prognosis of advanced HCC [8].

The incidence of HCC is associated with increased age, emphasizing the importance of comorbidity management for HCC since elderly patients present a higher prevalence of chronic comorbidities, including diabetes mellitus (DM) and hypertension (HTN) [9–11]. In 2017, there were 451 million diabetic patients globally and responded 9.9% of worldwide deaths [10]. Besides, HTN contributes to over 45% of heart-disease deaths, 51% of stroke-related deaths and 9.4 million total deaths worldwide annually [11]. A recent study proposed DM is associated with a longer time-to-progression in sorafenib-treated HCC patients [12]. However, some studies explored no survival difference between diabetic and nondiabetic HCC patients treated with sorafenib [13, 14]. Due to inconsistent results, whether DM affects the prognosis of sorafenib-treated HCC cannot be drawn to definite conclusion. In addition, the prognostic role of HTN remains unknown in sorafenib-treated HCC patients. Of note, the adverse health consequences of HTN are compounded since many patients possess risk factors, including obesity and DM, which increase the odds of heart attack, stroke and kidney failure [11]. Therefore, DM and HTN are closely linked [11]; when assessed, one cannot be confirmed to affect patients' survival without excluding the other. By grouping the study patients based on the presence of non HTN-associated DM, non DM-associated HTN and comorbid DM plus HTN, we could differentiate the individual roles of DM and HTN in sorafenib-treated advanced HCC. To attain greater clinical benefit, we also assessed the survival effects between different diabetes medications, including metformin, non-metformin oral hypoglycemic agents (OHA) and regular insulin (RI)/neutral protamine hagedorn (NPH).

## Materials and methods

### Patients and study design

During the period from August 2012 to February 2018, 733 HCC patients exhibiting Child-Pugh class A (score 5 or 6) with intra-hepatic venous invasion and/or extra-hepatic metastases accepted sorafenib therapy at China Medical University Hospital, Taichung, Taiwan. Based on their comorbidity at baseline (the presence/absence of DM or HTN), they were divided into four groups: control (patients without DM and HTN, n = 353), DM-only [patients with DM but without HTN (non HTN-associated DM), n = 91], HTN-only [patients with HTN but without DM (non DM-associated HTN), n = 184] and DM+HTN groups (patients with DM and HTN, n = 105). Furthermore, to assess the survival effects of diabetes medications, DM-only and DM+HTN groups were combined as the cohort of diabetic patients (DM cohort, n = 196) which was divided into three groups base on the types of given diabetes medications: metformin (n = 63), non-metformin OHA (n = 104) and RI/NPH groups (n = 29). We evaluated patients' overall survival (OS) and progression-free survival (PFS) until December 2018. OS was defined as the time from starting sorafenib therapy to death or last follow-up. PFS was measured from the initiation of sorafenib therapy to the presence of progression disease (PD), death or last follow-up. Afterward, we compared the survival outcomes of separate groups (control vs. DM-only; control vs. HTN-only; control vs. DM+HTN; control vs. metformin; control vs. non-metformin OHA; control vs. RI/NPH).

### Sorafenib therapy: Administration and therapeutic response evaluation

In this study, standard administration of sorafenib was 400 mg twice daily (800 mg per day). Dosage reduction was considered if intolerable sorafenib-induced adverse events occurred. At each visit during the study period, patients accepted detailed history taking and physical examination. Following the criteria of National Health Insurance Administration in Taiwan, patients underwent therapeutic response evaluation every two to three months since baseline, and only those with benign response [complete response (CR), partial response (PR) or stable disease (SD) rated by modified Response Evaluation Criteria in Solid Tumors (mRECIST)] and the maintenance of Child-Pugh class A were allowed to continue the next course of sorafenib therapy. Besides, none of the enrolled patients received second-line treatment for HCC during the study period. To assess patients' response to sorafenib therapy, contrast-enhanced tomography, serum biochemical tests and scoring of Child-Pugh scale were performed every two to three months since the initiation of treatment.

### Statistical analysis

All statistical analyses were performed with SPSS 20.0 (Released 2011. IBM SPSS Statistics for Macintosh, Version 20.0. Armonk, NY: IBM Corp.). Categorical variables were presented as absolute frequencies with relative proportions and compared by using Fisher exact test. Baseline continuous data were expressed as mean ± standard deviation with range and compared by using independent-samples $t$-test. Data for the duration of sorafenib therapy were shown as median with range and compared by using Mann-Whitney U test. Survival analysis was performed with Kaplan-Meier method by which data of OS and PFS were shown as median ± standard error with 95% confidence interval (CI). The differences between survival curves were evaluated with Log-rank test. Hazard ratio (HR) was calculated with Cox regression model in which variables assessed under univariate analysis were all entered into multivariate analysis to confirm the correlation between explanatory and response variables. With the use of multivariate Cox regression, survival risks in experimental groups were adjusted by variables showing statistical significance in

homogeneity analysis of baseline characteristics. All statistical tests were two-tailed, and a p-value below 0.05 was considered statistically significant.

### Ethics statements

The present study was approved by the institutional review board of China Medical University Hospital (No. CMUH109-REC2-033). All procedures performed in the present study were in accordance with the ethical standards of the institutional review board and the 1964 Helsinki Declaration with its later amendments. Written informed consent was obtained from all individual participants.

## Results

### Patient characteristics

Patient characteristics in separate groups of the entire study cohort (n = 733) are shown in Table 1. At baseline, all enrolled HCC patients were Child-Pugh class A with intra-hepatic venous invasion and/or extra-hepatic metastases. Compared with control group, DM-only, HTN-only and DM+HTN groups had a significantly higher mean age at baseline (control vs. DM-only, 57.67±12.24 vs. 63.12±9.42 years, p<0.001; control vs. HTN-only, 57.67±12.24 vs. 66.83±11.27 years, p<0.001; control vs. DM+HTN, 57.67±12.24 vs. 66.22±9.21 years, p<0.001) (Table 1), explaining the age-dependent prevalence of DM and HTN [10, 11].

### Survival outcomes

For the survival outcomes of control, DM-only, HTN-only and DM+HTN groups, median OS and PFS are shown in Table 2 with OS and PFS curves illustrated in Fig 1A and 1B respectively. Among these four groups, both DM-only and DM+HTN groups significantly presented better OS and PFS than control group (OS: control vs. DM-only, 7.70±0.58 vs. 11.83±1.38 months, p = 0.003; control vs. DM+HTN, 7.70±0.58 vs. 11.43±2.24 months, p = 0.008) (Table 2; Fig 1A) (PFS: control vs. DM-only, 3.70±0.37 vs. 6.83±1.70 months, p = 0.008; control vs. DM +HTN, 3.70±0.37 vs. 6.33±1.78 months, p = 0.004) (Table 2; Fig 1B). However, there was no significant survival difference between control and HTN-only groups (OS: 7.70±0.58 vs. 8.80 ±0.85 months, p = 0.111) (Table 2; Fig 1A) (PFS: 3.70±0.37 vs. 4.43±0.80 months, p = 0.094) (Table 2; Fig 1B).

### Survival risks

Table 3 shows respective HR of OS and PFS in DM-only, HTN-only or DM+HTN group compared with control group. To eliminate the bias, HR was adjusted by variables showing statistical significance in homogeneity analysis of baseline characteristics (homogeneity analysis: referred to Table 1). Compared with control group, the risk regarding death or PD was significantly lower in DM-only [OS: HR = 0.708 (95% CI: 0.547–0.917), p = 0.009; PFS: HR = 0.725 (95% CI: 0.560–0.939), p = 0.015] or DM+HTN group [OS: HR = 0.720 (95% CI: 0.560–0.925), p = 0.010; PFS: HR = 0.668 (95% CI: 0.519–0.859), p = 0.002] (Table 3). Besides, in comparison with control group, HTN-only group showed an insignificant difference of risk regarding death [OS: HR = 0.845 (95% CI: 0.687–1.039), p = 0.110] or PD [PFS: HR = 0.841 (95% CI: 0.683–1.037), p = 0.106] (Table 3).

### Variables associated with survival in the entire study cohort

Survival risks under different categorizing variables in the entire study cohort are shown in Tables 4 and 5. Among all variables, three factors significantly correlated with better OS in

**Table 1. Patient characteristics in separate groups of the entire study cohort (n = 733).**

| | Separate groups of the entire study cohort (n = 733)[a] | | | | p-value | | |
| --- | --- | --- | --- | --- | --- | --- | --- |
| | I: Control (n = 353) | II: DM-only (n = 91) | III: HTN-only (n = 184) | IV: DM+HTN (n = 105) | I vs. II | I vs. III | I vs. IV |
| **Baseline characteristics** | | | | | | | |
| Male† | 307 (87.0%) | 71 (78.0%) | 140 (76.1%) | 71 (67.6%) | 0.046* | 0.002* | <0.001* |
| Age, mean (range)‡ | 57.67±12.24 (21–85) | 63.12±9.42 (36–84) | 66.83±11.27 (33–88) | 66.22±9.21 (44–85) | <0.001* | <0.001* | <0.001* |
| HBV or HCV infection† | | | | | | | |
| HBV only | 161 (45.6%) | 41 (45.1%) | 61 (33.2%) | 26 (24.8%) | 1.000 | 0.006* | <0.001* |
| HCV only | 27 (7.6%) | 11 (12.1%) | 30 (16.3%) | 10 (9.5%) | 0.206 | 0.003* | 0.543 |
| HBV+HCV | 8 (2.3%) | 2 (2.2%) | 3 (1.6%) | 1 (1.0%) | 1.000 | 0.756 | 0.691 |
| None | 157 (44.5%) | 37 (40.7%) | 90 (48.9%) | 68 (64.8%) | 0.554 | 0.362 | <0.001* |
| Liver cirrhosis† | 285 (80.7%) | 76 (83.5%) | 148 (80.4%) | 88 (83.8%) | 0.651 | 1.000 | 0.568 |
| Tumor site† | | | | | | | |
| Intra-hepatic venous invasion only | 149 (42.2%) | 39 (42.9%) | 67 (36.4%) | 37 (35.2%) | 0.906 | 0.228 | 0.215 |
| Extra-hepatic metastases only | 166 (47.0%) | 44 (48.4%) | 102 (55.4%) | 55 (52.4%) | 0.906 | 0.069 | 0.374 |
| lymph nodes | 32 (19.3%) | 11 (25.0%) | 25 (24.5%) | 9 (16.4%) | 0.406 | 0.357 | 0.694 |
| lung | 69 (41.6%) | 20 (45.5%) | 40 (39.2%) | 21 (38.2%) | 0.732 | 0.798 | 0.752 |
| adrenal gland | 6 (3.6%) | 1 (2.3%) | 3 (2.9%) | 1 (1.8%) | 1.000 | 1.000 | 0.684 |
| bone | 23 (13.9%) | 5 (11.4%) | 15 (14.7%) | 7 (12.7%) | 0.806 | 0.858 | 1.000 |
| other[b] | 15 (9.0%) | 1 (2.3%) | 4 (3.9%) | 8 (14.5%) | 0.202 | 0.144 | 0.307 |
| multi-organ | 21 (12.7%) | 6 (13.6%) | 15 (14.7%) | 9 (16.4%) | 0.805 | 0.713 | 0.499 |
| Intra-hepatic venous invasion plus extra-hepatic metastases | 38 (10.8%) | 8 (8.8%) | 15 (8.2%) | 13 (12.4%) | 0.701 | 0.364 | 0.601 |
| BP (mmHg)[c]‡ | | | | | | | |
| Systolic, mean (range) | 123.38±12.46 (90–164) | 127.40±15.10 (87–160) | 140.66±19.03 (104–206) | 137.95±18.67 (103–190) | 0.021* | <0.001* | <0.001* |
| Diastolic, mean (range) | 76.54±9.08 (50–107) | 75.19±9.56 (53–99) | 80.49±12.82 (10–115) | 76.97±11.35 (44–114) | 0.211 | <0.001* | 0.720 |
| Glucose, mean (mg/dL) (range)[c]‡ | 116.16±27.38 (25–289) | 168.73±62.01 (73–433) | 122.01±27.17 (74–214) | 167.28±65.19 (66–443) | <0.001* | 0.019* | <0.001* |
| AFP, mean (ng/mL) (range)‡ | 8529.92±17138.68 (1.48–54001.00) | 3135.10±9459.73 (0.91–54001.00) | 6257.92±14091.10 (0.89–54001.00) | 3767.92±11294.73 (1.16–54001.00) | <0.001* | 0.101 | 0.001* |
| **Sorafenib duration, median (month) (range)[d]§** | 2.67 (0.10–66.63) | 4.00 (0.13–61.40) | 3.63 (0.10–73.97) | 4.13 (0.17–77.63) | 0.034* | 0.014* | 0.006* |
| **Events during the study†** | | | | | | | |
| Expired | 315 (89.2%) | 78 (85.7%) | 154 (83.7%) | 90 (85.7%) | 0.358 | 0.076 | 0.384 |
| PD | 237 (67.1%) | 56 (61.5%) | 138 (75.0%) | 68 (64.8%) | 0.323 | 0.061 | 0.640 |

Abbreviation: DM, diabetes mellitus; HTN, hypertension; HBV, hepatitis B virus; HCV, hepatitis C virus; BP, blood pressure; AFP, alpha-fetoprotein; PD, progression disease.

[a]At baseline, all patients were Child-Pugh class A with intra-hepatic venous invasion and/or extra-hepatic metastases.

[b]Tumor involvement of an extra-hepatic organ other than lymph nodes, lung, adrenal gland and bone.

[c]To eliminate the bias, data of baseline blood pressure (BP) and glucose level for each patient were determined by calculating the mean values of BP and glucose level measured multiple (two or three) times at baseline.

[d]The duration of sorafenib therapy depended on the therapeutic response evaluated every two to three months since baseline. Patients allowed to continue the next course of sorafenib therapy were those with benign response [complete response (CR), partial response (PR) or stable disease (SD) rated by modified Response Evaluation Criteria in Solid Tumors (mRECIST)] and the maintenance of Child-Pugh class A.

†Fisher exact test.

‡independent-samples *t*-test.

§Mann-Whitney U test.

*A p-value below 0.05 was considered statistically significant.

**Table 2. Overall survival (OS) and progression-free survival (PFS) in separate groups of the entire study cohort (n = 733).**

|  | OS, median (95% CI) (month)† | PFS, median (95% CI) (month)† |
|---|---|---|
| **Control (n = 353)** | 7.70±0.58 (6.57–8.84) | 3.70±0.37 (2.98–4.43) |
| **DM-only (n = 91)** | 11.83±1.38 (9.14–14.53) | 6.83±1.70 (3.50–10.17) |
| **HTN-only (n = 184)** | 8.80±0.85 (7.14–10.46) | 4.43±0.80 (2.86–6.01) |
| **DM+HTN (n = 105)** | 11.43±2.24 (7.05–15.82) | 6.33±1.78 (2.84–9.82) |
| | Log-rank test | |
| | **OS** | **PFS** |
| **Control vs. DM-only** | p = 0.003* | p = 0.008* |
| **Control vs. HTN-only** | p = 0.111 | p = 0.094 |
| **Control vs. DM+HTN** | p = 0.008* | p = 0.004* |

Abbreviation: OS, overall survival; CI, confidence interval; PFS, progression-free survival; DM, diabetes mellitus; HTN, hypertension.

†Kaplan-Meier method: OS and PFS were shown as median ± standard error with 95% CI.

*A p-value below 0.05 was considered statistically significant.

univariate analysis, including the presence of DM (HR = 0.748, p = 0.001), hepatitis B virus (HBV) and/or hepatitis C virus (HCV) infection (HR = 0.714, p<0.001) and baseline alpha-fetoprotein (AFP) <400 ng/mL (HR = 0.584, p<0.001) (Table 4). Furthermore, all of these factors independently predicted better OS in multivariate analysis (DM: HR = 0.762, p = 0.003; HBV and/or HCV infection: HR = 0.709, p<0.001; baseline AFP <400 ng/mL: HR = 0.598, p<0.001) (Table 4). Conversely, intra-hepatic venous invasion and multi-organ metastases significantly predicted poorer OS in univariate (HR = 1.230, p = 0.009; HR = 1.352, p = 0.033 respectively) and multivariate analysis (HR = 1.251, p = 0.011; HR = 1.431, p = 0.015 respectively) (Table 4). Of note, the presence of HTN insignificantly correlated with OS in univariate (HR = 0.872, p = 0.094) and multivariate analysis (HR = 0.906, p = 0.251) (Table 4).

For the predictors of PFS, four factors significantly correlated with better PFS in univariate analysis, including the presence of DM (HR = 0.747, p = 0.001), HTN (HR = 0.852, p = 0.049), HBV and/or HCV infection (HR = 0.672, p<0.001) and baseline AFP <400 ng/mL (HR = 0.611, p<0.001) (Table 5). Among these factors, those independently predicting better PFS in multivariate analysis were the presence of DM (HR = 0.770, p = 0.005), HBV and/or HCV infection (HR = 0.655, p<0.001) and baseline AFP <400 ng/mL (HR = 0.616, p<0.001), while the presence of HTN was insignificantly associated with PFS in multivariate analysis (HR = 0.865, p = 0.093) (Table 5). At last, multi-organ metastases (HR = 1.448, p = 0.009) and intra-hepatic venous invasion plus extra-hepatic metastases (HR = 1.331, p = 0.026) significantly predicted poorer PFS in univariate analysis; in multivariate analysis, the former remained statistically significant (HR = 1.516, p = 0.005) while the later was insignificant (HR = 1.238, p = 0.129) (Table 5).

## The prognostic roles of diabetes medications

Patient characteristics in separate groups of the DM cohort are shown in S1 Table. Among the DM cohort, baseline or on-sorafenib (during sorafenib therapy) hemoglobin A1c (HbA1c) level <7% (considered well-controlled) insignificantly correlated with the duration of sorafenib therapy in univariate (baseline: HR = 0.953, p = 0.755; on-sorafenib: HR = 0.865, p = 0.348) and multivariate analysis (baseline: HR = 0.979, p = 0.908; on-sorafenib: HR = 0.810, p = 0.254) (S2 Table). For the survival outcomes in separate groups of the DM cohort, median

**A**

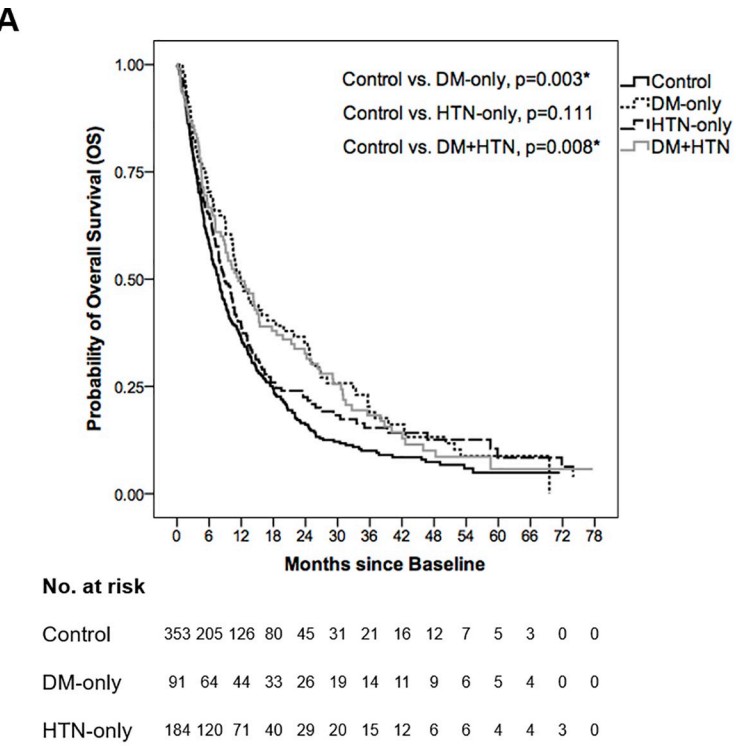

No. at risk

| | | | | | | | | | | | | | |
|---|---|---|---|---|---|---|---|---|---|---|---|---|---|
| Control | 353 | 205 | 126 | 80 | 45 | 31 | 21 | 16 | 12 | 7 | 5 | 3 | 0 | 0 |
| DM-only | 91 | 64 | 44 | 33 | 26 | 19 | 14 | 11 | 9 | 6 | 5 | 4 | 0 | 0 |
| HTN-only | 184 | 120 | 71 | 40 | 29 | 20 | 15 | 12 | 6 | 6 | 4 | 4 | 3 | 0 |
| DM+HTN | 105 | 70 | 52 | 38 | 29 | 21 | 15 | 10 | 7 | 3 | 2 | 2 | 2 | 0 |

**B**

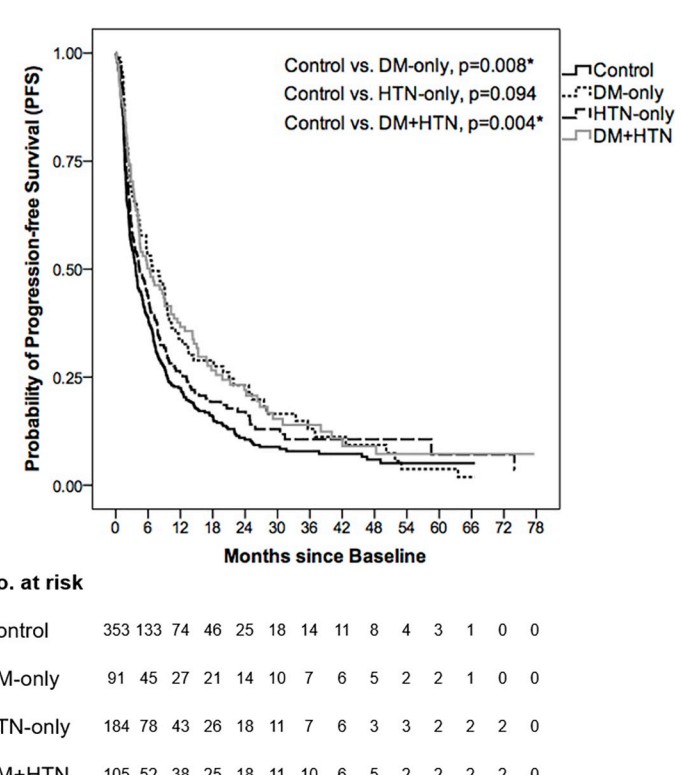

No. at risk

| | | | | | | | | | | | | | |
|---|---|---|---|---|---|---|---|---|---|---|---|---|---|---|
| Control | 353 | 133 | 74 | 46 | 25 | 18 | 14 | 11 | 8 | 4 | 3 | 1 | 0 | 0 |
| DM-only | 91 | 45 | 27 | 21 | 14 | 10 | 7 | 6 | 5 | 2 | 2 | 1 | 0 | 0 |
| HTN-only | 184 | 78 | 43 | 26 | 18 | 11 | 7 | 6 | 3 | 3 | 2 | 2 | 2 | 0 |
| DM+HTN | 105 | 52 | 38 | 25 | 18 | 11 | 10 | 6 | 5 | 2 | 2 | 2 | 2 | 0 |

**Fig 1. Overall survival (OS) and progression-free survival (PFS) curves of control, DM-only, HTN-only and DM +HTN groups.** (A) The median OS was 7.70±0.58, 11.83±1.38, 8.80±0.85 and 11.43±2.24 months in control, DM-only, HTN-only and DM+HTN groups respectively. In comparison with control group, both DM-only and DM+HTN groups had better OS (p = 0.003 and p = 0.008 respectively) while HTN-only group showed an insignificant OS difference (p = 0.111). (B) The median PFS was 3.70±0.37, 6.83±1.70, 4.43±0.80 and 6.33±1.78 months in control, DM-only, HTN-only and DM+HTN groups respectively. In comparison with control group, both DM-only and DM+HTN groups had better PFS (p = 0.008 and p = 0.004 respectively) while HTN-only group showed an insignificant PFS difference (p = 0.094). *Log-rank test: A p-value below 0.05 was considered statistically significant.

OS and PFS are shown in Table 6 with OS and PFS curves illustrated in Fig 2A and 2B respectively. In comparison with control group, all separate groups of the DM cohort, including metformin, non-metformin OHA and RI/NPH groups, significantly presented better OS (control vs. metformin, 7.70±0.58 vs. 12.60±2.17 months, p = 0.011; control vs. non-metformin OHA, 7.70±0.58 vs. 10.80±1.20 months, p = 0.016; control vs. RI/NPH, 7.70±0.58 vs. 15.20±4.45 months, p = 0.026) (Table 6; Fig 2A) and PFS (control vs. metformin, 3.70±0.37 vs. 8.17±1.53 months, p = 0.009; control vs. non-metformin OHA, 3.70±0.37 vs. 5.67±1.57 months, p = 0.017; control vs. RI/NPH, 3.70±0.37 vs. 7.17±2.04 months, p = 0.039) (Table 6; Fig 2B).

## Discussion

Since DM is closely linked with HCC development at both molecular [15, 16] and epidemiological levels [15–18], we previously presumed DM could lead to poorer prognosis in advanced HCC patients receiving sorafenib. However, our study found that DM predicts better OS and PFS in sorafenib-treated advanced HCC patients. To explain the unexpected results, we inferred specific mechanisms may lead to better prognosis in DM-associated advanced HCC treated with sorafenib. Besides, diabetes medications with anti-tumor effects may co-contribute to the positive prognostic role of DM in advanced HCC treated with sorafenib.

**Table 3. Survival risks in DM-only (n = 91), HTN-only (n = 184) or DM+HTN group (n = 105) compared with control group (n = 353).**

|  | OS | | PFS | |
|---|---|---|---|---|
|  | **HR (95% CI)** | **p-value** | **HR (95% CI)** | **p-value** |
| **Control (n = 353)** | Ref | | Ref | |
| **DM-only (n = 91)** | 0.708 (0.547–0.917)† | 0.009* | 0.725 (0.560–0.939)† | 0.015* |
| **HTN-only (n = 184)** | 0.845 (0.687–1.039)‡ | 0.110 | 0.841 (0.683–1.037)‡ | 0.106 |
| **DM+HTN (n = 105)** | 0.720 (0.560–0.925)§ | 0.010* | 0.668 (0.519–0.859)§ | 0.002* |

Abbreviation: OS, overall survival; PFS, progression-free survival; HR, hazard ratio; CI, confidence interval; Ref, reference variable; DM, diabetes mellitus; HTN, hypertension.

HR was adjusted by variables showing statistical significance in homogeneity analysis of baseline characteristics (referred to Table 1).

†adjusted by male frequency, age, systolic blood pressure (BP) and alpha-fetoprotein (AFP) level at baseline. (Glucose level was excluded from adjustment since DM-only group contained diabetic patients.).

‡adjusted by male frequency, age, percentage of cases with hepatitis B virus (HBV) infection only, percentage of cases with hepatitis C virus (HCV) infection only and glucose level at baseline. (Systolic and diastolic BP were excluded from adjustment since HTN-only group contained HTN patients.).

§adjusted by male frequency, age, percentage of cases with HBV infection only, percentage of cases without HBV/ HCV infection and AFP level at baseline. (Systolic BP and glucose level were excluded from adjustment since DM +HTN group contained patients with DM plus HTN.).

*Cox regression model: A p-value below 0.05 was considered statistically significant.

**Table 4. Cox regression of overall survival (OS) in the entire study cohort (n = 733).**

| variable | case number | Univariate analysis | | Multivariate analysis | |
|---|---|---|---|---|---|
| | | HR (95% CI) | p-value | HR (95% CI) | p-value |
| **Baseline characteristics** | | | | | |
| DM(+)/HTN(+ or -) | | | | | |
| No | 537 | Ref | | Ref | |
| Yes | 196 | 0.748 (0.627–0.894) | 0.001* | 0.762 (0.636–0.913) | 0.003* |
| HTN(+)/DM(+ or -) | | | | | |
| No | 444 | Ref | | Ref | |
| Yes | 289 | 0.872 (0.743–1.023) | 0.094 | 0.906 (0.765–1.072) | 0.251 |
| Sex | | | | | |
| Female | 144 | Ref | | Ref | |
| Male | 589 | 1.111 (0.911–1.354) | 0.299 | 1.155 (0.941–1.420) | 0.169 |
| Age | | | | | |
| <65 | 409 | Ref | | Ref | |
| ≥65 | 324 | 1.079 (0.923–1.262) | 0.341 | 1.152 (0.974–1.363) | 0.098 |
| HBV and/or HCV infection | | | | | |
| No | 352 | Ref | | Ref | |
| Yes | 381 | 0.714 (0.611–0.835) | <0.001* | 0.709 (0.604–0.831) | <0.001* |
| Liver cirrhosis | | | | | |
| No | 136 | Ref | | Ref | |
| Yes | 597 | 0.848 (0.695–1.034) | 0.102 | 0.921 (0.753–1.126) | 0.422 |
| Intra-hepatic venous invasion | | | | | |
| No | 367 | Ref | | Ref | |
| Yes | 366 | 1.230 (1.053–1.437) | 0.009* | 1.251 (1.052–1.486) | 0.011* |
| Multi-organ metastases | | | | | |
| No | 672 | Ref | | Ref | |
| Yes | 61 | 1.352 (1.025–1.784) | 0.033* | 1.431 (1.072–1.912) | 0.015* |
| Intra-hepatic venous invasion plus extra-hepatic metastases | | | | | |
| No | 659 | Ref | | Ref | |
| Yes | 74 | 1.191 (0.926–1.531) | 0.174 | 0.997 (0.759–1.311) | 0.986 |
| AFP | | | | | |
| ≥400 ng/mL | 289 | Ref | | Ref | |
| <400 ng/mL | 444 | 0.584 (0.498–0.685) | <0.001* | 0.598 (0.509–0.704) | <0.001* |

Abbreviation: HR, hazard ratio; CI, confidence interval; DM, diabetes mellitus; Ref, reference variable; HTN, hypertension; HBV, hepatitis B virus; HCV, hepatitis C virus; AFP, alpha-fetoprotein. To confirm the correlation between each variable and OS, all variables were entered into multivariate analysis.

*A p-value below 0.05 was considered statistically significant.

Though insulin-resistance-related hyperinsulinemia and DM-related chronic inflammation promote HCC development, specific DM-associated mechanisms, including reduction of hepatic glycolysis and impairment of insulin hypersecretion, may exert anti-tumor effects in sorafenib-treated HCC. Wang *et al.* proposed hepatic gluconeogenesis is significantly reduced in HCC via interleukin (IL)-6-Stat3-mediated activation of microRNA-23a which suppresses glucose-6-phosphatase and the transcription factor PGC-1a, aiding HCC growth and proliferation by maintaining a high level of glycolysis required for cancerous cells [19]. Besides, Tesori *et al.* reported that gene expression of HCC cells shifts toward glycolysis, diminishing sorafenib cytotoxicity which can be strengthened by glycolysis inhibition [20]. Furthermore, hepatic glycolysis is reduced under DM status due to insulin resistance [21]. These findings collectively

**Table 5. Cox regression of progression-free survival (PFS) in the entire study cohort (n = 733).**

| variable | case number | Univariate analysis | | Multivariate analysis | |
|---|---|---|---|---|---|
| | | HR (95% CI) | p-value | HR (95% CI) | p-value |
| **Baseline characteristics** | | | | | |
| DM(+)/HTN(+ or -) | | | | | |
|   No | 537 | Ref | | Ref | |
|   Yes | 196 | 0.747 (0.626–0.892) | 0.001* | 0.770 (0.642–0.923) | 0.005* |
| HTN(+)/DM(+ or -) | | | | | |
|   No | 444 | Ref | | Ref | |
|   Yes | 289 | 0.852 (0.726–1.000) | 0.049* | 0.865 (0.729–1.025) | 0.093 |
| Sex | | | | | |
|   Female | 144 | Ref | | Ref | |
|   Male | 589 | 1.135 (0.931–1.384) | 0.211 | 1.170 (0.954–1.436) | 0.132 |
| Age | | | | | |
|   <65 | 409 | Ref | | Ref | |
|   ≥65 | 324 | 1.069 (0.914–1.251) | 0.402 | 1.153 (0.973–1.365) | 0.100 |
| HBV and/or HCV infection | | | | | |
|   No | 352 | Ref | | Ref | |
|   Yes | 381 | 0.672 (0.575–0.785) | <0.001* | 0.655 (0.559–0.768) | <0.001* |
| Liver cirrhosis | | | | | |
|   No | 136 | Ref | | Ref | |
|   Yes | 597 | 0.832 (0.682–1.014) | 0.069 | 0.886 (0.725–1.082) | 0.236 |
| Intra-hepatic venous invasion | | | | | |
|   No | 367 | Ref | | Ref | |
|   Yes | 366 | 1.085 (0.929–1.268) | 0.302 | 1.053 (0.887–1.249) | 0.558 |
| Multi-organ metastases | | | | | |
|   No | 672 | Ref | | Ref | |
|   Yes | 61 | 1.448 (1.097–1.910) | 0.009* | 1.516 (1.136–2.025) | 0.005* |
| Intra-hepatic venous invasion plus extra-hepatic metastases | | | | | |
|   No | 659 | Ref | | Ref | |
|   Yes | 74 | 1.331 (1.035–1.712) | 0.026* | 1.238 (0.940–1.630) | 0.129 |
| AFP | | | | | |
|   ≥400 ng/mL | 289 | Ref | | Ref | |
|   <400 ng/mL | 444 | 0.611 (0.521–0.716) | <0.001* | 0.616 (0.524–0.725) | <0.001* |

Abbreviation: HR, hazard ratio; CI, confidence interval; DM, diabetes mellitus; Ref, reference variable; HTN, hypertension; HBV, hepatitis B virus; HCV, hepatitis C virus; AFP, alpha-fetoprotein. To confirm the correlation between each variable and PFS, all variables were entered into multivariate analysis.

*A p-value below 0.05 was considered statistically significant.

suggest decreased glycolysis in hepatocytes suppresses HCC tumorigenesis and resistance to sorafenib, explaining the positive prognostic role of DM in this study. On the other hand, pro-tumor hyperinsulinemia in type 2 DM is followed by hypoinsulinemia [22–24] due to β-cell dysfunction led by oxidative stress [23, 24], which diminishes the hyperinsulinemia-related negative effect of DM in HCC prognosis. Therefore, for DM-associated HCC, patients benefiting better from sorafenib may be those with less expressed hepatic glycolysis or reduced insulin hypersecretion.

In the present study, prescribed OHA included metformin, repaglinide, acarbose, glimepiride, glipizide, glibenclamide (glyburide), gliclazide, linagliptin, sitagliptin, vildagliptin, saxagliptin and pioglitazone. Metformin, the first-line OHA for type 2 DM, is found to directly

**Table 6. Overall survival (OS) and progression-free survival (PFS) in control group (patients without DM and HTN; n = 353) and separate groups of the DM cohort (diabetic patients with or without HTN, i.e. the combination cohort of DM-only and DM+HTN groups; n = 196).**

| | OS, median (95% CI) (month)† | PFS, median (95% CI) (month)† |
|---|---|---|
| **Control (n = 353)** | 7.70±0.58 (6.57–8.84) | 3.70±0.37 (2.98–4.43) |
| **Separate groups of the DM cohort (n = 196): divided by the types of diabetes medications** | | |
| **Metformin (n = 63)** | 12.60±2.17 (8.34–16.86) | 8.17±1.53 (5.16–11.17) |
| **Non-metformin OHA (n = 104)** | 10.80±1.20 (8.44–13.16) | 5.67±1.57 (2.59–8.75) |
| **RI/NPH (n = 29)** | 15.20±4.45 (6.49–23.91) | 7.17±2.04 (3.17–11.16) |
| | **Log-rank test** | |
| | **OS** | **PFS** |
| **Control vs. Metformin** | p = 0.011* | p = 0.009* |
| **Control vs. Non-metformin OHA** | p = 0.016* | p = 0.017* |
| **Control vs. RI/NPH** | p = 0.026* | p = 0.039* |

Abbreviation: OS, overall survival; CI, confidence interval; PFS, progression-free survival; DM, diabetes mellitus; OHA, oral hypoglycemic agents; RI, regular insulin; NPH, neutral protamine hagedorn.

†Kaplan-Meier method: OS and PFS were shown as median ± standard error with 95% CI.

*A p-value below 0.05 was considered statistically significant.

impacts on tumors by activating adenosine 5'-monophosphate-activated protein kinase (AMPK), leading to the suppression of mammalian Target of Rapamycin (mTOR) signaling pathway [25–27] (Fig 3), thus inhibiting tumor growth, survival and proliferation [28] (Fig 3). In addition, metformin indirectly exerts an anti-tumor effect via reducing serum insulin level, repressing the signaling pathway of insulin and insulin-like growth factor 1 (I/IGF-1)/phos-phoinositide 3-kinase (PI3K)/Akt/mTOR [25–27] or I/IGF-1/Ras/Raf/MEK/ERK [29, 30] (Fig 3). Several epidemiologic and clinical studies have reported a lower risk of HCC development [25–27, 31, 32] or better HCC prognosis [33, 34] with metformin use. On the contrary, some studies proposed metformin worsens sorafenib-treated HCC patients' survival since metformin shares overlapping anti-tumor mechanisms with sorafenib (Fig 3), making the resistance to metformin induces a poorer response to sorafenib [13, 14]. However, in the present study, there was a better survival in sorafenib/metformin-treated patients compared with control group (patients without DM and HTN) (OS: 12.60 vs. 7.70 months, p = 0.011; PFS: 8.17 vs. 3.70 months, p = 0.009) (Table 6). In addition, the duration of sorafenib therapy was considered as an indicator of treatment response since only patients with CR, PR or SD rated by mRECIST were allowed to continue sorafenib therapy in this study. The result showed that sorafenib/metformin-treated patients underwent longer median duration of sorafenib therapy (i.e. better response to sorafenib) than control group (5.00 vs. 2.67 months, p = 0.003), indicating metformin-induced resistance to sorafenib did not present in our study.

In addition to metformin, several non-metformin OHA are as well reported to exert anti-tumor effects, including repaglinide, glipizide, glibenclamide, sitagliptin, vildagliptin and pio-glitazone. Firstly, repaglinide possesses cytotoxic effects against hepatic, breast and cervical carcinoma cells (*HepG2*, *MCF-7* and *HeLa* cells) [35], or reduces the expression of Bcl-2, Beclin-1 and PD-L1 in glioma tissues [36]. Secondly, glipizide inhibits endothelial cell migration and tubular formation via up-regulating the expression of natriuretic peptide receptor A to suppress tumor angiogenesis [37, 38]. Thirdly, glibenclamide significantly induces HCC cell apoptosis via activating reactive-oxygen-species-dependent JNK pathway [39] and arrests HCC growth [40]. Fourthly, sitagliptin and vildagliptin trigger the infiltration of natural killer cells and T-cells into xenograft or liver tumors in rodent models [41]. Furthermore,

**A**

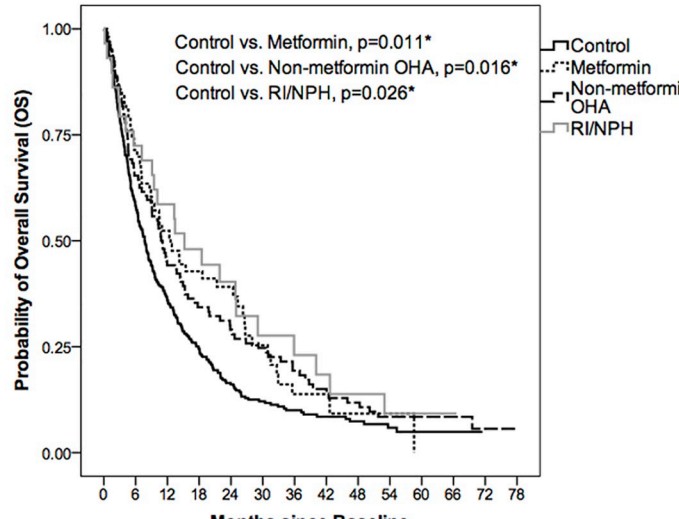

No. at risk

|  | | | | | | | | | | | | | |
|---|---|---|---|---|---|---|---|---|---|---|---|---|---|
| Control | 353 | 205 | 126 | 80 | 45 | 31 | 21 | 16 | 12 | 7 | 5 | 3 | 0 | 0 |
| Met | 63 | 45 | 33 | 25 | 18 | 11 | 6 | 3 | 2 | 1 | 0 | 0 | 0 | 0 |
| Non-met OHA | 104 | 68 | 46 | 33 | 27 | 23 | 18 | 14 | 11 | 6 | 5 | 4 | 2 | 0 |
| RI/NPH | 29 | 21 | 17 | 13 | 10 | 6 | 5 | 4 | 3 | 2 | 2 | 2 | 0 | 0 |

Abbreviation: Met, metformin.

**B**

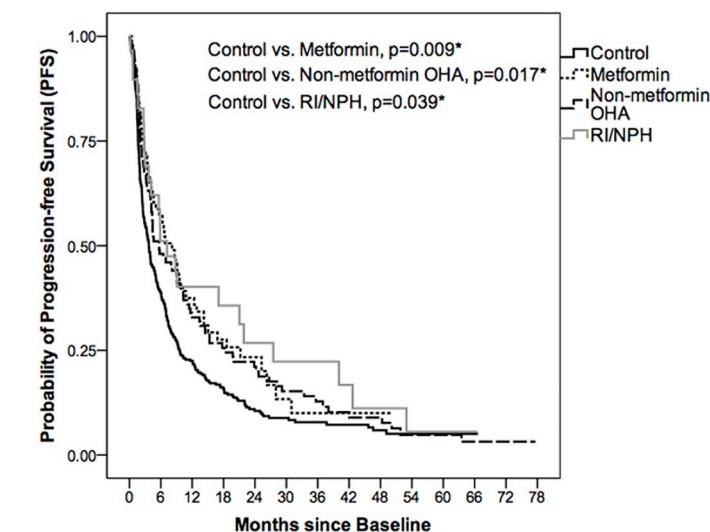

No. at risk

|  | | | | | | | | | | | | | |
|---|---|---|---|---|---|---|---|---|---|---|---|---|---|---|
| Control | 353 | 133 | 74 | 46 | 25 | 18 | 14 | 11 | 8 | 4 | 3 | 1 | 0 | 0 |
| Met | 63 | 35 | 23 | 15 | 8 | 4 | 3 | 1 | 1 | 0 | 0 | 0 | 0 | 0 |
| Non-met OHA | 104 | 48 | 32 | 23 | 18 | 13 | 10 | 8 | 7 | 3 | 3 | 2 | 2 | 0 |
| RI/NPH | 29 | 14 | 10 | 8 | 6 | 4 | 4 | 3 | 2 | 1 | 1 | 1 | 0 | 0 |

Abbreviation: Met, metformin.

**Fig 2. Overall survival (OS) and progression-free survival (PFS) curves of control, metformin, non-metformin OHA and RI/NPH groups.** (A) The median OS was 7.70±0.58, 12.60±2.17, 10.80±1.20 and 15.20±4.45 months in control, metformin, non-metformin OHA and RI/NPH groups respectively. In comparison with control group, metformin, non-metformin OHA and RI/NPH groups significantly presented better OS (p = 0.011, p = 0.016 and p = 0.026 respectively). (B) The median PFS was 3.70±0.37, 8.17±1.53, 5.67±1.57 and 7.17±2.04 months in control, metformin, non-metformin OHA and RI/NPH groups respectively. Compared with control group, metformin, non-metformin OHA and RI/NPH groups significantly presented better PFS (p = 0.009, p = 0.017 and p = 0.039 respectively). *Log-rank test: A p-value below 0.05 was considered statistically significant.

vildagliptin prevents the angiogenesis of high-fat-diet-induced HCC via down-regulating the dipeptidyl peptidase-4/chemokine ligand 2/angiogenesis pathway [42]. At last, pioglitazone increases circulating adiponectin production which activates hepatic AMPK signaling and down-regulates the mitogen-activated protein kinase pathways (mainly ERK/JNK/cJUN), thus inhibiting HCC development in cirrhotic rodent models [43].

In addition to direct anti-tumor mechanisms, the anti-inflammatory effects of OHA also exert indirect anti-tumor efficacy. Several agents have been shown to effectively decrease the

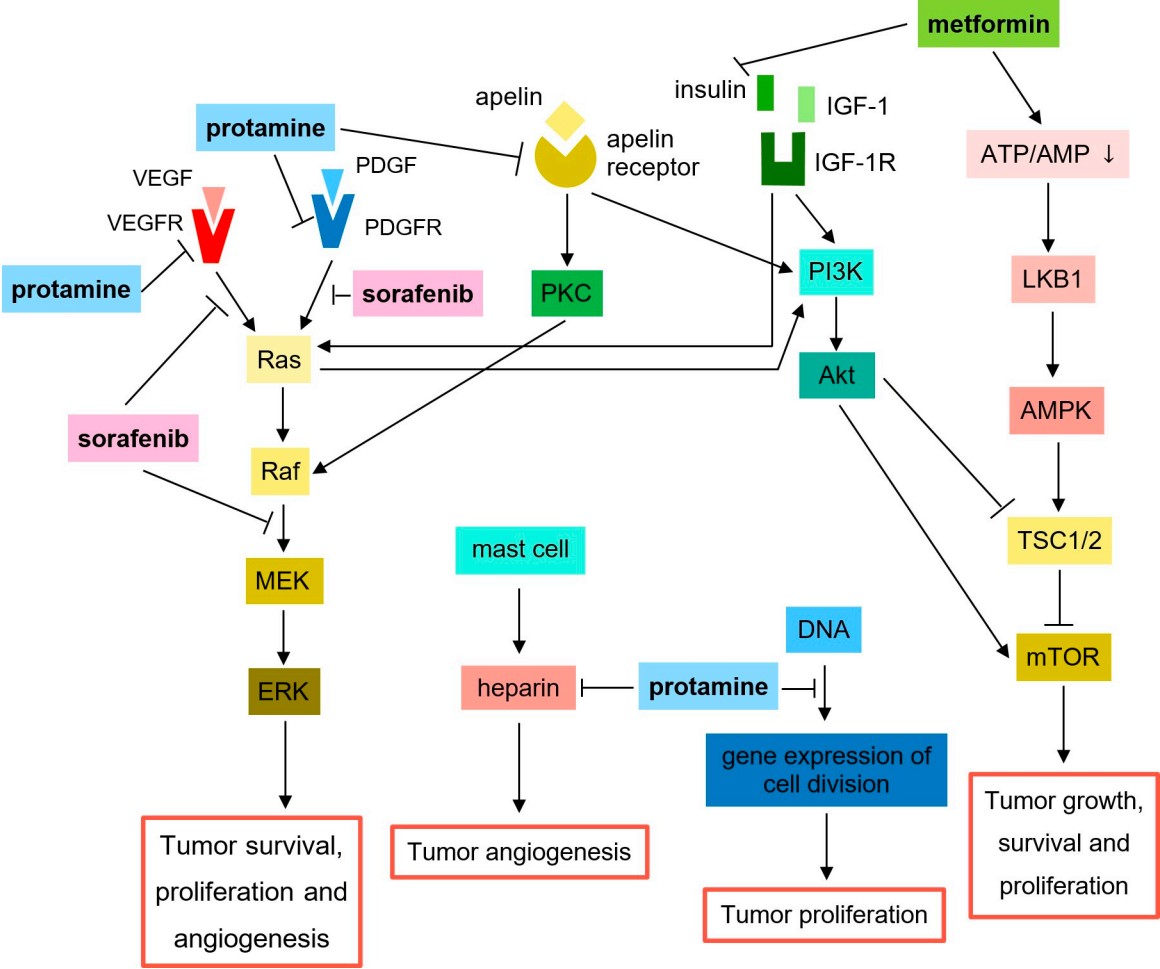

**Fig 3. Anti-tumor mechanisms of sorafenib, metformin and protamine: A review of previous studies [7, 8, 14, 25–30, 59–66].** Abbreviation: VEGF, vascular endothelial growth factor; VEGFR, VEGF receptor; PDGF, platelet-derived growth factor; PDGFR, PDGF receptor; PKC, protein kinase C; IGF-1, insulin-like growth factor 1; IGF-1R, IGF-1 receptor; PI3K, phosphoinositide 3-kinase; LKB1, liver kinase B1; AMPK, adenosine 5'-monophosphate-activated protein kinase; TSC1/2, tuberous sclerosis proteins 1 and 2 complex; mTOR, mammalian Target of Rapamycin.

levels of tumor necrosis factor (TNF)-α and IL-6, including glimepiride, repaglinide [44], acarbose [45–47], glibenclamide [48], gliclazide [49], linagliptin [50, 51] and saxagliptin [52–54]. TNF-α, a pro-inflammatory cytokine, significantly over-expresses in HCC patients and favors inflammation leading to poorer prognosis in sorafenib-treated HCC [55, 56]. Besides, TNF-α induces epithelial-mesenchymal transition which stimulates HCC proliferation, invasion and resistance to sorafenib [57]. As for IL-6, we previously demonstrated it expresses positive correlation with HCC severity via hepatocarcinogenesis-activating p-STAT3 pathway [55, 58].

RI/NPH mainly contains the ingredients of insulin and protamine, and its anti-tumor effects may derive from protamine instead of insulin due to HCC-inducing exogenous hyperinsulinemia [15, 16]. Protamine, an arginine-rich protein, performs anti-tumor effects through various mechanisms (Fig 3). During tumor angiogenesis, mast cells accumulate at the edges of tumors and initiate ingrowth of new capillary sprouts by releasing heparin, a sulfated glycosaminoglycan which facilitates the migration of capillary endothelial cells toward tumor sites [59]. Protamine binds to heparin and represses its stimulation of capillary endothelial cell migration or induces thrombosis of tumor vessels via neutralizing the anti-coagulant effect of heparin [59, 60] (Fig 3). In addition, protamine inhibits tumor angiogenesis via attaching to VEGFR [61, 62], PDGFR [61, 63] and apelin receptor [64, 65] (Fig 3). Protamine is also effective in arresting the proliferation of two fast-growing cell systems (*E. coli* and *HeLa* cells) due to the high binding affinity of arginine-rich protamine to negatively charged DNA backbones which leads to the transcriptional stop of gene expression related to cell division [66].

Sorafenib stimulates blood pressure (BP) elevation via VEGF signaling pathway (VSP) inhibition [67], and BP is a valid pharmacodynamic biomarker of VSP inhibition [68]. Therefore, sorafenib-induced HTN is associated with better prognosis in sorafenib-treated HCC patients [69]. However, it remains unknown whether non sorafenib-induced HTN affects the survival of advanced HCC patients receiving sorafenib. In this study, we found that baseline HTN showed an insignificant impact on patients' survival, indicating different prognostic roles between sorafenib-induced and non sorafenib-induced HTN. Furthermore, non HTN-associated DM and DM/HTN comorbidity at baseline are both linked with better prognosis, implying HTN does not eliminate the positive role of DM in sorafenib-treated advanced HCC.

In the present study, independent predictors of better OS and PFS also included HBV and/or HCV infection and baseline AFP <400 ng/mL. Non HBV and/or HCV-related tumors may lead to poorer survival in HCC patients due to delayed cancer detection [70]. Besides, baseline AFP level at a cut-off point of 400 ng/mL predicts long-term survival in unselected HCC patients [71] as our study showed lower level (<400 ng/mL) benefits survival in sorafenib-treated patients. On the contrary, liver cirrhosis played an insignificantly prognostic role, which may be attributed to all enrolled patients belonging to Child-Pugh class A and thus sharing a similar status of liver function at baseline. In addition, intra-hepatic venous invasion was independently associated with shorter OS, and multi-organ metastases independently predicted poorer OS and PFS. Yada *et al.* indicated that the hepatic arterial system supplies intra-hepatic tumors with abundant blood flow, making these lesions less likely to be affected by the anti-angiogenic effect of sorafenib in comparison with metastatic tumors requiring intensive angiogenesis to acquire sufficient blood flow [72]. Collectively, these findings imply sorafenib may exert better efficacy in patients with mono-organ metastasis compared to those with multi-organ metastases or intra-hepatic venous invasion.

Our study was unable to confirm which of the diabetes medications indicated better prognosis in the study patients due to statistical insignificance (S3–S5 Tables). Future studies are suggested to enroll more cases to solve this limitation. Nonetheless, our study identified that the prescription of metformin, non-metformin OHA and RI/NPH are all associated with prolonged survival and thus out of prognostic concern in DM-related advanced HCC patients

receiving sorafenib. Among these patients, baseline or on-sorafenib HbA1c level <7% insignificantly correlated with the duration of sorafenib therapy (an indicator of treatment response) in our study, implying well-controlled serum HbA1c level contributes limitedly to improving diabetic patients' response to sorafenib therapy.

## Conclusions

For sorafenib-treated advanced HCC, DM is associated with better prognosis probably due to specific mechanisms and diabetes medications including metformin, non-metformin OHA and RI/NPH. Besides, the prognostic efficacy of sorafenib is independent of baseline HTN in advanced HCC.

## Supporting information

**S1 File. Patient dataset of the present study.**
(XLSX)

**S1 Table. Patient characteristics in each separate group of the DM cohort (diabetic patients with or without HTN, i.e. the combination cohort of DM-only and DM+HTN groups; n = 196).**
(PDF)

**S2 Table. Assessing the correlation between hemoglobin A1c (HbA1c) level and sorafenib duration (an indicator of treatment response) in the DM cohort (diabetic patients with or without HTN, i.e. the combination cohort of DM-only and DM+HTN groups; n = 196): A Cox regression model.**
(PDF)

**S3 Table. Survival differences among separate groups of the DM cohort (diabetic patients with or without HTN, i.e. the combination cohort of DM-only and DM+HTN groups; n = 196).**
(PDF)

**S4 Table. Cox regression of overall survival (OS) in the DM cohort (diabetic patients with or without HTN, i.e. the combination cohort of DM-only and DM+HTN groups; n = 196).**
(PDF)

**S5 Table. Cox regression of progression-free survival (PFS) in the DM cohort (diabetic patients with or without HTN, i.e. the combination cohort of DM-only and DM+HTN groups; n = 196).**
(PDF)

## Acknowledgments

The authors are sincerely grateful to all patients participating in the present study. The authors thank Dr. Shu-Mei Tsai for her assistance in applying for the ethics approval of the institutional review board. The first author Ming-Han Hsieh extends his sincere gratitude to Dr. Chun-Hsiung Hsieh and Ms. Tsui-Huang Tsai for their encouragement in conducting the present study.

## Author Contributions

**Conceptualization:** Ming-Han Hsieh, Jung-Ta Kao.

**Data curation:** Ming-Han Hsieh, Tzu-Yu Kao, Ting-Hui Hsieh, Chun-Chi Kao, Jung-Ta Kao.

**Formal analysis:** Ming-Han Hsieh, Tzu-Yu Kao, Ting-Hui Hsieh, Chun-Chi Kao.

**Investigation:** Ming-Han Hsieh, Tzu-Yu Kao, Ting-Hui Hsieh, Chun-Chi Kao, Jung-Ta Kao.

**Methodology:** Ming-Han Hsieh, Jung-Ta Kao.

**Project administration:** Jung-Ta Kao.

**Resources:** Cheng-Yuan Peng, Hsueh-Chou Lai, Po-Heng Chuang, Jung-Ta Kao.

**Software:** Ming-Han Hsieh, Tzu-Yu Kao, Ting-Hui Hsieh, Chun-Chi Kao.

**Supervision:** Jung-Ta Kao.

**Validation:** Ming-Han Hsieh, Jung-Ta Kao.

**Visualization:** Ming-Han Hsieh.

**Writing – original draft:** Ming-Han Hsieh.

**Writing – review & editing:** Jung-Ta Kao.

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
