## [Decision Letter · Decision Letter 0]

20 Oct 2020

PONE-D-20-28054

Prognostic roles of pretreatment diabetes mellitus and hypertension in advanced hepatocellular carcinoma treated with sorafenib

PLOS ONE

Dear Dr. Kao,

Thank you for submitting your manuscript to PLOS ONE. After careful consideration, we feel that it has merit but does not fully meet PLOS ONE’s publication criteria as it currently stands. Therefore, we invite you to submit a revised version of the manuscript that addresses the points raised during the review process.

Please consult a statistician for re-evaluating the statistical analysis and an English-native speaker for the revised manuscript.

We look forward to receiving your revised manuscript.

Kind regards,

Ming-Lung Yu, MD, PhD

Academic Editor

PLOS ONE

Journal Requirements:

2. In ethics statement in the manuscript and in the online submission form, please provide additional information about the patient records/samples used in your retrospective study.

Specifically, please ensure that you have discussed whether all data/samples were fully anonymized before you accessed them and/or whether the IRB or ethics committee waived the requirement for informed consent.

If patients provided informed written consent to have data/samples from their medical records used in research, please include this information.

Reviewers' comments:

Reviewer's Responses to Questions

**Comments to the Author**

1. Is the manuscript technically sound, and do the data support the conclusions?

Reviewer #1: Yes

Reviewer #2: Partly

2. Has the statistical analysis been performed appropriately and rigorously? 

Reviewer #1: Yes

Reviewer #2: No

3. Have the authors made all data underlying the findings in their manuscript fully available?

Reviewer #1: Yes

Reviewer #2: Yes

4. Is the manuscript presented in an intelligible fashion and written in standard English?

Reviewer #1: Yes

Reviewer #2: Yes

5. Review Comments to the Author

Reviewer #1: It is an important issue to explore the prognosis factors in advanced HCC patients receiving sorafenib treatment. In this study, Hsieh et al explored whether DM and hypertension were associated with better prognosis in 733 patients with advanced HCC after sorafenib treatment. They found the presence of DM was associated with better prognosis counterintuitively. They also found survival benefits in different DM medication when compared to non-DM group.

This is an interesting finding although the result is opposite to what we have expected. I have the following comments

Specific comments

1. The authors found patients receiving different diabetes medication were all associated with better prognosis when compared to those without diabetes. As the authors did not identify any survival benefits among patients with different medication, I don’t think the benefits comes from the medication. Instead, the authors should discuss the possibility of different carcinogenesis mechanism of diabetes-related HCC, which may benefit better from sorafenib?

2. As this is an issue of DM being a prognostic factor, please explore whether HbA1c level is associated with treatment response, especially in those patients with diabetes.

3. Was there anyone receiving 2nd-line treatment after sorafenib failed.

Reviewer #2: Dear Editors:

In this manuscript, Hsieh and colleagues retrospectively analyzed data collected from 733 patients with advanced hepatocellular carcinoma (HCC) treated with sorafenib to investigate the association of diabetes mellitus (DM) with overall survival (OS) and progression-free survival (PFS). The authors reported that DM was associated with a lower risk of OS and PFS in both univariable and multivariable analyses. They further showed the association was consistent among diabetic patients grouped by anti-diabetes medication.

Overall, this study dealt with a clinically relevant topic with scientific novelty. The findings were thought-provocative and might add to our knowledge on the management of HCC with potential implications in tumor biology if they were confirmed. Nonetheless, there are a few concerns about the methodology, those regarding the analysis in particular. Specific comments are listed below.

Major:

1. Given that diabetic patients and the controls were unsurprisingly different in many baseline characteristics, it was a daunting challenge to affirm the association was not confounded. The authors resorted to multivariable modeling but there are quite a few pitfalls. For instance, only variables with a p value below 0.05 in the univariable analysis were examined. Response variable (duration of sorafenib which indicated the PFS and likely OS as well) was considered as an explanatory variable. Besides, continuous variable (alfa fetoprotein) was dichotomized without a reason. I’d suggest the authors consult a statistic expert at least for variable selection for the modelling.

2. From the introductory statements, it is unclear why the authors chose to group patients according to hypertension as well. This factorial design might incur unnecessary concerns of multiple testing and statistical power.

3. It is inappropriate to dogmatically claim causal relationship in the association. A statement like “metformin, non-metformin OHA and RI/NPH all exert a survival benefit” is unacceptably misleading.

Minor:

1. The term “pretreatment” DM/HTN was ambiguous. It might be interpreted as the status prior to treatment for DM/HTN (i.e., “pretreatment” served as an adjective for DM/HTN).

6. PLOS authors have the option to publish the peer review history of their article (what does this mean?). If published, this will include your full peer review and any attached files.

Reviewer #1: No

Reviewer #2: No

---

## [Author Response · Author response to Decision Letter 0]

22 Nov 2020

Academic Editor

Comment:

Please consult a statistician for re-evaluating the statistical analysis and an English-native speaker for the revised manuscript. 

Response:

Thank you very much for your suggestion. We have consulted a statistician to re-evaluate the statistical methods and an English-native speaker to revise the manuscript.

Reviewer #1

Comment:

It is an important issue to explore the prognosis factors in advanced HCC patients receiving sorafenib treatment. In this study, Hsieh et al explored whether DM and hypertension were associated with better prognosis in 733 patients with advanced HCC after sorafenib treatment. They found the presence of DM was associated with better prognosis counterintuitively. They also found survival benefits in different DM medication when compared to non-DM group.

This is an interesting finding although the result is opposite to what we have expected. I have the following comments. 

Response:

Thank you very much for your appreciation and compliment. We have followed your comments and revised our manuscript.

Comment:

1. The authors found patients receiving different diabetes medication were all associated with better prognosis when compared to those without diabetes. As the authors did not identify any survival benefits among patients with different medication, I don’t think the benefits comes from the medication. Instead, the authors should discuss the possibility of different carcinogenesis mechanism of diabetes-related HCC, which may benefit better from sorafenib? 

Response:

Thank you for your comments. Our study cannot confirm which diabetes medication benefits prognosis the most. To explain the positive role of DM more properly, we have added the possible mechanisms in discussion part according to reviewer’s suggestion. The revisions contain the following points.

In the unmarked version of the revised manuscript:

1. Page 21 line 292-296: ‘‘To explain the unexpected results, we inferred specific mechanisms may lead to better prognosis in DM-associated advanced HCC treated with sorafenib. Besides, diabetes medications with anti-tumor effects may co-contribute to the positive prognostic role of DM in advanced HCC treated with sorafenib’’.

2. Page 22 line 297-315: ‘‘Though insulin-resistance-related hyperinsulinemia and DM-related chronic inflammation promote HCC development, specific DM-associated mechanisms, including reduction of hepatic glycolysis and impairment of insulin hypersecretion, may exert anti-tumor effects in sorafenib-treated HCC. Wang et al. proposed hepatic gluconeogenesis is significantly reduced in HCC via interleukin (IL)-6-Stat3-mediated activation of microRNA-23a which suppresses glucose-6-phosphatase and the transcription factor PGC-1a, aiding HCC growth and proliferation by maintaining a high level of glycolysis required for cancerous cells [19]. Besides, Tesori et al. reported that gene expression of HCC cells shifts toward glycolysis, diminishing sorafenib cytotoxicity which can be strengthened by glycolysis inhibition [20]. Furthermore, hepatic glycolysis is reduced under DM status due to insulin resistance [21]. These findings collectively suggest decreased glycolysis in hepatocytes suppresses HCC tumorigenesis and resistance to sorafenib, explaining the positive prognostic role of DM in this study. On the other hand, pro-tumor hyperinsulinemia in type 2 DM is followed by hypoinsulinemia [22-24] due to β-cell dysfunction led by oxidative stress [23,24], which diminishes the hyperinsulinemia-related negative effect of DM in HCC prognosis. Therefore, for DM-associated HCC, patients benefiting better from sorafenib may be those with less expressed hepatic glycolysis or reduced insulin hypersecretion’’.

Comment:

2. As this is an issue of DM being a prognostic factor, please explore whether HbA1c level is associated with treatment response, especially in those patients with diabetes.

Response:

Thank you very much for your comment. We have added the assessment of correlation between HbA1c level and sorafenib treatment response in our manuscript.

In the unmarked version of the revised manuscript:

1. Page 19 line 255-259: ‘‘Among the DM cohort, baseline or on-sorafenib (during sorafenib therapy) hemoglobin A1c (HbA1c) level <7 % (considered well-controlled) insignificantly correlated with the duration of sorafenib therapy in univariate (baseline: HR=0.953, p=0.755; on-sorafenib: HR=0.865, p=0.348) and multivariate analysis (baseline: HR=0.979, p=0.908; on-sorafenib: HR=0.810, p=0.254) (S2 Table)’’.

2. Page 23 line 333-336: ‘‘In addition, the duration of sorafenib therapy was considered as an indicator of treatment response since only patients with CR, PR or SD rated by mRECIST were allowed to continue sorafenib therapy in this study’’.

3. Page 28 line 421-425: ‘‘Among these patients, baseline or on-sorafenib HbA1c level <7 % insignificantly correlated with the duration of sorafenib therapy (an indicator of treatment response) in our study, implying well-controlled serum HbA1c level contributes limitedly to improving diabetic patients’ response to sorafenib therapy’’.

4. Page 38 line 656-659: ‘‘S2 Table. Assessing the correlation between hemoglobin A1c (HbA1c) level and sorafenib duration (an indicator of treatment response) in the DM cohort (diabetic patients with or without HTN, i.e. the combination cohort of DM-only and DM+HTN groups; n=196): a Cox regression model’’.

Comment:

3. Was there anyone receiving 2nd-line treatment after sorafenib failed. 

Response:

Thank you for your comments. In this study, none of the enrolled patients received second-line treatment for HCC. We have added this statement in our manuscript.

(Unmarked version of the revised manuscript page 7 line 122-123 added ‘‘Besides, none of the enrolled patients received second-line treatment for HCC during the study period’’).

Reviewer #2

Comment:

In this manuscript, Hsieh and colleagues retrospectively analyzed data collected from 733 patients with advanced hepatocellular carcinoma (HCC) treated with sorafenib to investigate the association of diabetes mellitus (DM) with overall survival (OS) and progression-free survival (PFS). The authors reported that DM was associated with a lower risk of OS and PFS in both univariable and multivariable analyses. They further showed the association was consistent among diabetic patients grouped by anti-diabetes medication.

Overall, this study dealt with a clinically relevant topic with scientific novelty. The findings were thought-provocative and might add to our knowledge on the management of HCC with potential implications in tumor biology if they were confirmed. Nonetheless, there are a few concerns about the methodology, those regarding the analysis in particular. Specific comments are listed below. 

Response:

Thank you for your comments. According to your concerns, we have consulted statistical experts and revised our manuscript, especially (1) the modification of variable selection for multivariate analysis, (2) a more detailed explanation for grouping patients according to HTN and (3) the adjustment of inappropriate misleading statements.

Comment:

Major:

1. Given that diabetic patients and the controls were unsurprisingly different in many baseline characteristics, it was a daunting challenge to affirm the association was not confounded. The authors resorted to multivariable modeling but there are quite a few pitfalls. For instance, only variables with a p value below 0.05 in the univariable analysis were examined. Response variable (duration of sorafenib which indicated the PFS and likely OS as well) was considered as an explanatory variable. Besides, continuous variable (alfa fetoprotein) was dichotomized without a reason. I’d suggest the authors consult a statistic expert at least for variable selection for the modelling.

Response:

Thank you for your comments. We have consulted statistical experts to revise the statistical method of multivariate analysis performed in our study. Specific points are listed as bellow.

1. To avoid biased variable selection and confirm the correlation between explanatory variables and response variables, all variables assessed under univariate analysis were entered into multivariate analysis. This statistical method was adopted by previous study to validate the statistical correlation 

(El-Sherif O, Jiang ZG, Tapper EB, Huang KC, Zhong A, Osinusi A, et al. Baseline Factors Associated With Improvements in Decompensated Cirrhosis After Direct-Acting Antiviral Therapy for Hepatitis C Virus Infection. Gastroenterology. 2018;154(8):2111-2121.e8.).

In the unmarked version of the revised manuscript:

(1) Page 8 line 138-141: ‘‘Hazard ratio (HR) was calculated with Cox regression model in which variables assessed under univariate analysis were all entered into multivariate analysis to confirm the correlation between explanatory and response variables’’.

(2) Table 4-5, S2 Table and S4-S5 Table have all been revised accordingly.

(3) Major study results remain unchanged after revising the statistical methods.

2. We have deleted the duration of sorafenib therapy from Cox regression model since it is a response variable instead of an explanatory variable. Thank you very much for helping us correct this mistake.

3. Since the threshold of alpha-fetoprotein (AFP) level with 400 ng/mL is proposed to predict HCC prognosis, we have modified the cut-off point of AFP level as <400 vs. ≥400 ng/mL and provided the related reference (Reference 71: Hsu C-Y, Liu P-H, Lee Y-H, Hsia C-Y, Huang Y-H, Lin H-C, et al. Using Serum α-Fetoprotein for Prognostic Prediction in Patients with Hepatocellular Carcinoma: What is the Most Optimal Cutoff? PLOS ONE. 2015;10(3):e0118825.).

Comment:

2. From the introductory statements, it is unclear why the authors chose to group patients according to hypertension as well. This factorial design might incur unnecessary concerns of multiple testing and statistical power.

Response:

1. Thank you for your concern. We have added a more detailed explanation why we chose to group patients according to hypertension as well.

(Unmarked version of the revised manuscript page 5 line 77-85: ‘‘In addition, the prognostic role of HTN remains unknown in sorafenib-treated HCC patients. Of note, the adverse health consequences of HTN are compounded since many patients possess risk factors, including obesity and DM, which increase the odds of heart attack, stroke and kidney failure [11]. Therefore, DM and HTN are closely linked [11]; when assessed, one cannot be confirmed to affect patients’ survival without excluding the other. By grouping the study patients based on the presence of non HTN-associated DM, non DM-associated HTN and comorbid DM plus HTN, we could differentiate the individual roles of DM and HTN in sorafenib-treated advanced HCC’’). 

We believe this factorial design is important. Without this design, it may be questionable whether the prognostic role of DM is contributed or impaired by comorbid HTN.

2. To avoid the concern of multiple testing and statistical power, variables entered into Cox regression model were the two factors [DM (yes/no) and HTN (yes/no)] (Table 4-5) instead of the other three factors [DM without HTN (yes/no), HTN without DM (yes/no) and DM/HTN comorbidity (yes/no)].

Comment:

3. It is inappropriate to dogmatically claim causal relationship in the association. A statement like “metformin, non-metformin OHA and RI/NPH all exert a survival benefit” is unacceptably misleading.

Response:

Thank you for your comment. We have screened the entire manuscript and deleted every statement dogmatically claiming causal relationship in the association.

In the unmarked version of the revised manuscript:

1. Page 3 line 44-45: The statement “Besides, metformin, non-metformin OHA and RI/NPH all exert a survival benefit” has been revised as ‘‘Besides, metformin, non-metformin OHA and RI/NPH are associated with longer survival among DM-related advanced HCC patients receiving sorafenib’’.

2. Page 19 line 253: The title of this section has been revised as ‘‘The prognostic roles of diabetes medications’’.

3. Page 28 line 418-421: ‘‘Nonetheless, our study identified that the prescription of metformin, non-metformin OHA and RI/NPH are all associated with prolonged survival and thus out of prognostic concern in DM-related advanced HCC patients receiving sorafenib’’.

4. Page 28 line 428-430: ‘‘For sorafenib-treated advanced HCC, DM is associated with better prognosis probably due to specific mechanisms and diabetes medications including metformin, non-metformin OHA and RI/NPH’’.

Comment:

Minor:

1. The term “pretreatment” DM/HTN was ambiguous. It might be interpreted as the status prior to treatment for DM/HTN (i.e., “pretreatment” served as an adjective for DM/HTN).

Response:

Thank you for your comment. We have deleted the term ‘‘pretreatment’’ which is no longer shown in the manuscript. The full manuscript title now reads ‘‘Prognostic roles of diabetes mellitus and hypertension in advanced hepatocellular carcinoma treated with sorafenib’’ and the short title now reads ‘‘DM and HTN in sorafenib-treated advanced HCC’’.

Abbreviation: HCC, hepatocellular carcinoma; DM, diabetes mellitus; HTN, hypertension.

---

## [Decision Letter · Decision Letter 1]

8 Dec 2020

Prognostic roles of diabetes mellitus and hypertension in advanced hepatocellular carcinoma treated with sorafenib

PONE-D-20-28054R1

Dear Dr. Kao,

We’re pleased to inform you that your manuscript has been judged scientifically suitable for publication and will be formally accepted for publication once it meets all outstanding technical requirements.

Kind regards,

Ming-Lung Yu, MD, PhD

Academic Editor

PLOS ONE

Additional Editor Comments (optional):

Reviewers' comments:

Reviewer's Responses to Questions

**Comments to the Author**

1. If the authors have adequately addressed your comments raised in a previous round of review and you feel that this manuscript is now acceptable for publication, you may indicate that here to bypass the “Comments to the Author” section, enter your conflict of interest statement in the “Confidential to Editor” section, and submit your "Accept" recommendation.

Reviewer #1: All comments have been addressed

Reviewer #2: All comments have been addressed

2. Is the manuscript technically sound, and do the data support the conclusions?

Reviewer #1: Yes

Reviewer #2: Yes

3. Has the statistical analysis been performed appropriately and rigorously? 

Reviewer #1: Yes

Reviewer #2: Yes

4. Have the authors made all data underlying the findings in their manuscript fully available?

Reviewer #1: Yes

Reviewer #2: Yes

5. Is the manuscript presented in an intelligible fashion and written in standard English?

Reviewer #1: Yes

Reviewer #2: Yes

6. Review Comments to the Author

Reviewer #1: All the questions have been answered clearly. I have no more comments about this revised manuscript.

Reviewer #2: I thank the authors for taking the efforts to respond to my comments raised in the first round of review.

7. PLOS authors have the option to publish the peer review history of their article (what does this mean?). If published, this will include your full peer review and any attached files.

Reviewer #1: No

Reviewer #2: No

---

## [Editor Report · Acceptance letter]

22 Dec 2020

PONE-D-20-28054R1 

Prognostic roles of diabetes mellitus and hypertension in advanced hepatocellular carcinoma treated with sorafenib 

Dear Dr. Kao:

I'm pleased to inform you that your manuscript has been deemed suitable for publication in PLOS ONE. Congratulations! Your manuscript is now with our production department. 

Kind regards, 

on behalf of

Dr. Ming-Lung Yu 

Academic Editor

PLOS ONE